# Two Puzzles, a Tour Guide, and a Teacher: The First Cohorts’ Lived Experience of Participating in the MClSc Interprofessional Pain Management Program

**DOI:** 10.3390/healthcare11101397

**Published:** 2023-05-11

**Authors:** Zoe A. Leyland, David M. Walton, Elizabeth Anne Kinsella

**Affiliations:** 1Graduate Program in Health Rehabilitation Sciences, Faculty of Health Sciences, Western University, London, ON N6A 3K7, Canada; dwalton5@uwo.ca (D.M.W.);; 2School of Physical Therapy, Western University, London, ON N6C 1H1, Canada; 3Institute of Health Sciences Education, Faculty of Medicine and Health Sciences, McGill University, Montreal, QC H3A 1A3, Canada

**Keywords:** pain, pain education, interprofessional, lived-experience, reflection, master’s program

## Abstract

(1) Background: The Master of Clinical Science program (MClSc) in Advanced Healthcare Practice at (University) introduced a new “Interprofessional Pain Management” (IPM) field in September 2019. The purpose of this study is to inquire into the following research question: What are MClSc Interprofessional Pain Management students’ lived experiences of participating in pain management education? (2) This study followed an interpretivist research design. The text that was considered central to descriptions of the lived experience of participating in the IPM program was highlighted and organized into a spreadsheet and then sorted into themes. (3) Results: Five themes in regard to the lived experiences of participating in the first cohort of the MClSc IPM program were identified: Reflection on Stagnation in Professional Disciplines; Meaning Making Through Dialogue with Like-Minded Learners; Challenging Ideas and Critical Thinking at Play; Interprofessionalism as Part of Ideal Practice; and Becoming a Competent Person-Centred Partner in Pain Care. (4) Conclusions: This program offers a unique approach to learning while creating an online platform to work, collaborate, and challenge like-minded experts in the field of pain. In doing this research, we hope that more practitioners will work towards the goal of becoming competent, person-centered pain care providers.

## 1. Introduction

In September 2019, the Master of Clinical Science program (MClSc) in Advanced Healthcare Practice at (University) introduced a new “Interprofessional Pain Management” (IPM) field. Practicing healthcare providers with a special interest in pain were invited to apply. The first two authors started co-developing this program in September 2017 with guidance from experts, designing it around a collaborative team-integrated competencies framework. While there has been a lack of pain management education for all healthcare providers, a shift in educational philosophy, framework, and expectations is taking place, which has led to considerable innovation and challenges for healthcare professional educators [1]. This led to taking a competency-based curricula approach that focuses on the outcomes of learning and abilities rather than traditional time-based training.

There are five core competencies for the program, each with a set of entrustable professional activities (EPAs) that, when completed, provide evidence for adequate mastery. Details are presented in Table 1. Interprofessional education involving interactive, small-group learning formats and collaboration have been identified as key factors for effective pain education [2] and were included in the design of the program. Learners were also provided with numerous resources and access to content experts via video calls in various fields in relation to pain. As far as we know, this online and competency-based master’s degree program with an interprofessional emphasis is the first of its kind in Canada.

Understanding the lived experience of the first cohort is important for us as researchers to interpret participants’ learning journeys and experiences [3]. The findings can be used to improve and implement changes in the existing program and to inform other health professional education programs interested in advancing the design of pain education programs. Inadequate education of healthcare practitioners is a major and persistent barrier to safe and effective pain management for patients. This includes an absence of competency-based and interprofessional curricula [4]. This program focuses on improving clinical effectiveness in managing pain using learners’ real-world experiences as a platform for learning while taking a biopsychosocial approach, which was first proposed by George Engel in a paper published in 1977. This approach compromises the biomedical domain of organelles through cells, tissues, organs, and the nervous system to the phenomenological domain, which examines lived experiences of pain and pain management, to the psychosocial domain, which consists of subjective and emotional dimensions, as well as consideration of social and family structures (psychosocial domain) [5]. The purpose of this study is to inquire into the following research question: What are MClSc Interprofessional Pain Management students’ lived experiences of participating in pain management education?

## 2. Methods

### 2.1. Design

This study followed an interpretivist research design. Interpretive phenomenology, also referred to as hermeneutic phenomenology, aims to uncover the lived experiences of a person or several people, usually with a focus on a specific phenomenon. In this study, the phenomenon was the experience of participating in an interprofessional pain management program. We used van Manen’s approach to phenomenology of practice to inform the design of the study, including the construction of interview questions and the representation of findings as phenomenology that tries to show how our concepts, words, and theories may inevitably form and give structure to our experiences as we live them [6]. In using an interpretivist phenomenological approach, we follow the idea that phenomenology prioritizes how the human being *experiences* the world, such as how the patient or learner experiences pain [6]. Within this methodology, we inquired into participants’ experiences of participation in the course to better understand each learner’s perspective and story.

### 2.2. Theoretical Framework

Sharing stories has been described as fundamental to understanding and constructing meanings, knowledge, and identity [7]. In following a phenomenological framework, we also adopt a constructivist–interpretivist paradigmatic position to emphasize the aim of understanding the lived experiences of participants within social and historical contexts [8]. The phenomenon at hand is the experience of participating in an Interprofessional Pain Management master’s level program. The ultimate learning outcome for the IPM program is for learners to work towards becoming competent, person-centered pain care providers, which may be viewed as a narrative of the journey of participation in the program for the learners. The use of hermeneutical phenomenology enables the exploration of participants’ experiences with further abstraction and interpretation based on researchers’ theoretical and personal knowledge [3]. In order to understand the meaning of what is being communicated, especially when intentions, values, moral issues, and feelings are involved, this may require critical reflection of assumptions, which is also known as Transformative Learning Theory [9]. A strong background and understanding of pedagogy and curriculum from the first author helped form interpretation while engaging with the dialogue from participants through a critical lens and critical reflection for this study in understanding that learning is a mutually transformative relationship between work and identity. This transformation and reflective discourse are captured through the interviews. It is also important to mention that transformative learning has also been described as a critical reflection of assumptions that may occur either in a group or autonomously, and testing the validity of this may require critical-dialectical discourse [10], which is demonstrated in the analysis section of this research.

### 2.3. Participants

Participants were the first cohort of the IPM program recruited as a convenience sample. At an early stage, the learners were notified that they would be invited to participate in a study on their lived experience of participating in the program. The first cohort was made up of four learners: one chiropractor, one naturopath, and two physiotherapists. All of them were practicing at the time of the program and at the time of the interviews. Each learner was compensated for their participation through a stipend to compensate for their time away from clinical responsibilities.

### 2.4. Ethical Considerations

This study protocol was approved by the Research Ethics Board (REB) at Western University (project ID: 113244). Details of the study were explained to each participant before signing the consent form. As this is a small cohort/sample size, learners were notified through the letter of information and consent that there was a potential that participants would be revealed inadvertently by the data that they contributed, but no identifiers would be published. Code numbers were assigned to each participant ranging from P001–P004 and corresponded with the audio recording as well as the transcript for confidentiality and identity protection. In the consent form, the participants were informed that the primary researcher had been involved with the design and implementation of the IPM MClSc program since its inception as part of their degree requirements as PhD candidates. The primary researcher held an administrative role and served as a committee member to determine the policies and procedures of the program. They did not engage in delivering content and had no power over student progress in the IPM MClSc program.

### 2.5. Data Collection

A semi-structured interview guide was developed that aimed to elicit participants’ stories about their experience participating in the pain management program. A number of drafts of the guide were developed through iterative dialogue amongst the research team. The interview guide was piloted with the second author. (See Table 2 for the full interview guide). All interviews were conducted by the first author via Zoom. The interviews occurred after the completion of the program and after degrees were conferred. The method of question involved reflection, probing, and engagement in dialogue. The length of the interviews ranged from just less than 1 h and 45 min to just over 2 h.

### 2.6. Data Analysis

The selective reading approach developed by van Manen (2016) is used for data analysis. In selective reading, the text is read several times and explored to identify what is being revealed about the phenomenon being described through the perspective of the researcher [6]. The analysis also included creating a mind map using an online mind-mapping generator, which was shared with the second and third authors. See Figure 1 for Mind Map, which was used to demonstrate changing ideas over time, and was used to visually organize the findings into themes. According to van Manen (2016), phrases should be copied and then saved as possible rhetorical ‘gems’ to support the researcher while writing the phenomenological text [6]. The text that was considered central to descriptions of the lived experience of participating in the IPM program was highlighted by researchers, organized into a spreadsheet, and then sorted into themes. The authors engaged in transparent dialogue about the unfolding interpretations and considered if they could be interpreted otherwise but confirmed that the data and subsequent results were authentic and trustworthy. These themes were developed iteratively, with the final structure constructed through consensus of all authors and with the use of rigor, which consisted of peer debriefing and reflexive journaling.

## 3. Results

Five themes in regard to the lived experiences of participating in the first cohort of the MClSc IPM program were identified: Reflection on Stagnation in Professional Disciplines; Meaning Making Through Dialogue with Like-Minded Learners; Challenging Ideas and Critical Thinking at Play; Interprofessionalism as Part of Ideal Practice; and Becoming a Competent Person-Centred Partner in Pain Care.

### 3.1. Themes

#### 3.1.1. Theme 1. Reflection on Stagnation in Professional Disciplines

In asking participants why they applied to the Advanced Healthcare Practice MClSc IPM program, some learners expressed a feeling of stagnation in their current clinical role and spoke of looking for a way to challenge themselves.

“*Well, I thought as a sole practitioner, I felt like I was a little bit on an island in terms of keeping up-to-date and there’s only so much that I can do for continuing education, and I felt like that was getting a little bit stale*”—P003.

“*I’m extremely goal-oriented…this might be a fault to some extent. I always wanted something else. I’m always like, ‘OK, what’s next? How can I challenge myself?’ I never want to be stagnate in my career. So, this kind of came to be an option and…the pieces aligned and then it worked*”—P004.

Participants shared a desire to continue their education in pain management. The majority of them characterized themselves as ‘life-long learners’ and valued gaining new knowledge to better themselves as practitioners.

“*I was just getting very sort of frustrated with continuing education that the state of it in [clinical profession]. Anyway, I was really big into the soft tissue side of things…and I found that getting very stale and repetitive. You know, it’s almost like they weren’t keeping up with the current loop of research that I was reading on the internet, and I found that to be a little frustrating because if I’m gonna spend all this money, I want to do it just for something that is worthwhile and current*”—P003.

“*It’s important to me to become the best clinician I can. I thought this was a good opportunity to pursue becoming…an expert in pain*”—P001.

“*So, I think it was just sort of the hope for the growth and expanding on sort of my skills as a healthcare provider to sort of be continuing on that journey and grow there. I think one of my past sort of supervisors used to say that ‘as soon as you feel like you have nothing left to learn, it’s time to get out of the profession.’ And so, I think… I really just like learning and really enjoy learning, but I think it’s also, as a healthcare provider, we have this responsibility to keep growing and to not stop. So, I think part of my hope was just that I could use sort of where I’m coming from, but then also, the resource in this program to just sort of keep expanding on that and become a better healthcare provider for people that experience pain*”—P002.

Expressions of participants’ hopes and values as healthcare providers were woven through the interview conversations. 

#### 3.1.2. Theme 2. Meaning-Making through Dialogue with Like-Minded Learners

The participants’ peers in their cohort, mentors, and experts in various fields that the learners connected with throughout the program were frequently described as like-minded individuals. Dialogue with these individuals was described as intellectually stimulating and as informing rewarding learning opportunities.

“*I enjoyed the program because of my conversations with the people in it*”—P004.

“*I found it to be very stimulating intellectually and the other learners…and with anyone that we really talked to was very friendly and open with communication so, I found it to be something that I looked forward to. You know, it was a lot of work, but I enjoyed doing it and I would definitely do it again and I’d recommend it to anybody*”—P003.

“*We were able to have weekly or almost weekly meetings with the group. And so, I think the process of having that time for discussion to really break down the material, speak with different experts in their fields and learn from them, I think that was another key area that I think there’s a few other opportunities to really touch base with so many different experts on a weekly basis…that’s really challenging on your own. I don’t think we would have gotten that opportunity outside of this program to any significant extent, especially in such a small group where we could have those individual conversations with the expert. So, I think those pieces really stood out for me as it’s quite… [a] rewarding experience in being able to kind of really build working relationships with these other healthcare providers that you’re learning with and from at the same time*”—P002.

“*It is nice to just be…among other similar minded clinicians*”—P001.

Learning from and with others was highly valued and seemed to be a way to make sense of the newly acquired knowledge from the program. Participants explained the growth and connection that took place, and how meaning was made through dialogue with others.

“*I think some of the most positive things that I took…it’s definitely in terms of improving my own confidence with my own abilities. It’s really good to go through a rigorous program to kind of prove that…I do deserve to have a seat at the table, and I can have these meaningful conversations with other experts, so I think it’s been huge for my confidence as well*”—P001.

“*I think we have a tendency to focus on some of the harder or difficult moments because they really make an impact on us, but I think those are the ones that almost forced charge. And so, I think without some of those or part of the reason I wanted to do this program was the extra chance to be sort of mentored and…the competencies did [that] through that growth*”—P002.

“*I feel like the whole program itself was a bit of an ah-ha moment. But I think… a few of them for me were really sort of some of the papers that we kind of connected with themes throughout and how we sort of framework these experiences to sort of make sure that we can intervene in a way that makes sense for that person. So, I think some of the papers around the radar plot and sort of how do we integrate those within patient care? I think a lot of us do it to some extent, sort of intuitively, but then having that visual was sort of a bit of an ah-ha moment for me. It’s an easy way then to kind of explain to someone else of, ‘this is why we intervened here’ because this is the pain drivers*”—P002.

#### 3.1.3. Theme 3. Challenging Ideas and Critical Thinking at Play

Many participants described challenging their assumptions and deepening their critical thinking, and how their thought patterns had transformed through participation in the post-professional MClSc IPM program.

“*I didn’t feel like I was back in [clinical training] where you just kind of accept everything. I’m listening, but I’m challenging everything that they’re kind of suggesting and not just taking it as it is…I’m able to listen to some really great thinkers and I’m also able to still kind of challenge them*”—P001.

“*And sort of really almost challenging some of the beliefs that I had had about our interest, like the interventions that [clinical profession] use. And so, I think it was really in that sense, it was sort of stepping outside of my comfort zone because I have done some of that work in the past, but I don’t think to the same depth that I did it within this program in route of reassessing… It was really sort of that level of discomfort of grappling with…certain things and really sort of challenging my own comfort and beliefs about what we’re doing*”—P002.

In doing this critical thinking, many participants appeared to challenge their ideas and beliefs, while also thinking outside of the box.

“*I just assumed that a good recovery is you either get back to who you were or better and those are the only two options that were satisfactory, acceptable for me, but I think being able to really engage in that conversation and have that challenge, and also open up the options for people that there’s other ways that people can have good recoveries and I don’t need to tie my own…expectations of myself, and that’s not a reflection of myself…that was the first and biggest ah-ha moment or kind of paradigm shifting moment where I guess my perspective as a clinican changed*”—P001.

“*I like to question things and you know really make sure that what’s being said is valid*”—P004.

“*I remember there was a few [activities] that were definitely more intellectually hard to grasp [including] the Seven Step framework to a research paper and then also to the pain model. I remember those took quite a bit of time because I wasn’t sure what the expectations were so…I had to really think outside of the box and make sure I was looking at the big picture*”—P003.

Participants described learning activities and tools, such as portfolios, and how they provided evidence of transformation in thinking or understanding. Throughout this apparent transformation, participants described feeling empowered and accomplished.

“*[Portfolio] almost sort of reinforced what you’ve learned in one solid space, and so, having that opportunity to compile it all together was, I think, really quite powerful as a learner and quite a big [accomplishment] at the end of it*”—P002.

“*I found most of them [learning activities] to be really meaningful practices. So, I appreciated that, and I think it was nice to be able to create something…I guess there are some thoughts there that felt like I’m not doing this for myself anymore, whereas some of them felt deeply personal, that was such a wonderful thought experiment for me and the act of me going through it really was helpful for my own growth compared to some that felt like I’m just trying to prove to some external evaluator that I’m OK at something*”—P001.

“*I’m very proud of what I did. You know, I put a lot of work into it [portfolio]…I’d like to go back and read it, and I’m more than happy to let anyone have a look at it, but I definitely, towards the end of it, is when it really started to sink in that a lot of work went into that, so I felt really good about everything I had in there. You know some exercises were better than others, but everything in there I definitely feel good about*”—P003.

#### 3.1.4. Theme 4. The Spirit of Interprofessional Care includes Patient Partnership

The recognition of interprofessional care through learning with, and about each other, was demonstrated throughout the interviews. Participants recognized their own limitations within their practice contexts and challenged their current structures of care by reflecting on what their ideal practice would entail.

“*You don’t want the patient to feel like they’re in the middle of everyone and everyone’s sort of on their own little island doing their own thing. You know, even now, being in a solo practice, I know if I had a difficult patient, I could 100% trust that I could write anyone in our group [fellow cohort members] to get their opinion and I would value it and see what they had to say. And maybe it would make me think of something that I wasn’t doing or maybe that they could help them, or they knew somebody that could help this patient. So, I think that’s incredibly valuable*”—P003. 

“*How we approach things and then just even so much of how we can learn from each other and then how we can sort of synthesize all of that information that we’ve learned together to really work better together, and I think that’s one of the key pieces that I feel like I’ve learned a lot as well. Now, we have sort of the shared language of how we work together effectively. What does that structure look like and how can we best serve patients within that sort of interprofessional collaboration framework?*”—P002.

“*This was a consistent thing that came out that might come up with others as well, possibly, but something we had kind of spoken about is, most of us, we don’t work in super interdisciplinary teams…we’re very siloed and so our interaction with other professionals is so limited right now with where I am. I could say I’ve maybe been more aware of when we’ve kind of reached the limits of my abilities and we could be quicker to reach out to other people*”—P001.

“*My ideal practice would certainly involve some interprofessional collaboration*”—P003.

In interprofessional healthcare, person-centred approach is a way to demonstrate partnership and the therapeutic alliance with patients and caregivers. Participants discussed recognizing their own limitations and boundaries, as an important component of the learning process. Participants also shared their processes of advocating for patients; advocacy was identified as a value by the majority of participants.

“*I think it’s certainly a process that I still feel like will continue for myself just to sort of continue to grow and learn about as well. But I definitely think I feel like I do have those better understandings of my own boundaries and how I can even advocate for someone with about healthcare provider as well, and sort of merging that together…I mean you could come up with the perfect treatment plan or management strategies for someone, but if it doesn’t actually make sense for them and it’s not something that’s in line with their values, it’s pretty much useless…I think at the end of the days it’s always taking that patient or person first approach with them*”—P002.

“*I think the things that I really value is, huh, this might be outdates, but I think we really have a duty to do no harm and that really is a really broad thing that extends beyond just what we’d like to do with people. So, I think really making sure we are…trying to help people in a way that is most impactful for them both now and in the future. It’s really important. So, part of that means being mindful of your language, how you frame things, how you’re empowering people, and how you’re helping support the development of their own self-efficacy*”—P001.

“*I still think I have my interpersonal skills [that] I think are pretty good for the most part with most patients, so I can build, I would say, fairly consistent therapeutic alliances with people, which builds trust*”—P004.

Participants shared narratives about how they reassessed values or reflected on them more deeply through participating in the IPM program. Some spoke about how this kind of reflection and reassessment of values contributed to shifts in their identity, or how they see themselves as a person or practitioner.

“*Well, how I view myself, I find that I apply a lot of the same techniques and active listening. I can use that in my everyday life as well. So, again, I think I’m pretty easy-going and it certainly helped in interacting with people I feel overall. I can probably connect with people a little bit better and I’m comfortable doing that because we’ve had those experiences in those conversations. There was one about oppression and privilege, concepts that I hadn’t heard about, that I hadn’t really given too much actual thought to and then once we had those conversations, it certainly made me think about how I come across and putting myself in other people’s shoes. So, I think it’s helped shape my identity that way that I’m hopefully a better person because of that*”—P003.

“*I think my professional identity is really based in just sort of supporting people where they’re at and supporting their ability to have…access… to a management strategy that makes sense within the context of their values and belief systems…So, I really just kind of see it coming down to where I can support people within their value system and based on what sort of pain drivers are going on alongside these other healthcare providers as well and sot of always working together with that person at the centre of their care*”—P002.

“*Crazy ambitious goal, I do want to contribute to systemic change of how people with pain are cared for and I want healthcare to be more…evidence-based, more transparent…have patient and practitioner on an equal level…I much prefer to coach on the same level with people. So, I would love to see people with pain being more empowered and getting good care*”—P001.

#### 3.1.5. Theme 5. Becoming a Competent Person-Centred Partner in Pain Care

The importance of showing up and being present in the journey toward becoming a competent person-centred partner in pain care was shared by the participants. Some participants described their comfort level with pain management prior to starting the program and how the competencies in the program transformed their knowledge.

“*I think our competencies were a pretty good place to start, that those are things that people should embody, but I think you do need to be knowledgeable, there is a lot you need to do to be an expert in helping people with pain, and that’s hard to help people if you don’t even know the game that you’re playing*”—P001.

“*That’s the neat thing about competencies is that you can kind of come in at your own level of comfort and then sort of grow from there or demonstrate competency within that framework. So, I think that I had sort of like a mixed comfort with some of them. I don’t think I was a pain expert to any degree. I think that one is one that you sort of, as the field continues to grow or sort of expand and change, I think that is one that kind of keeps developing over time…some of the ones around empathic care or self-awareness and reflexivity, I think I came in with a bit of comfort around them already. Just sort of by way of my profession and training and sort of how we’ve sort of gone about that. Then pain research and interprofessional collaboration, a bit of experience there, but just sort of expanding on that throughout the program I think and kind of continuing to develop it*”—P002.

The knowledge gained from the program was described in detail throughout interviews, with all participants describing a change in how they practice.

“*Initially, I was a little unsure of exactly what it entailed, but you know shortly after we got started, I realized it would be an awesome addition to my clinical practice and I found myself applying what we were learning almost immediately, which was pretty amazing to do be able to do that. So, definitely it’s impacted the way that I practice*”—P003.

“*I would say I definitely feel like I’ve shifted and changed and grown as a healthcare provider in the field of pain management. I think that has certainly shifted, not only sort of my professional life, but I think it’s also sort of spilled over into my personal life as well. I think it’s sort of inevitable in a lot of ways of just when you’re focusing on certain areas…you know how I’ve gotten here and sort of what my biases are and sort of checking that I think it’s hard to capture in one or two sentences sort of how it’s impacted me, but I think it’s certainly made me a stronger healthcare provider, but also more aware in my personal life as well*”—P002.

“*I think this program…to be honest…maybe [was an] expensive lesson, but it kind of reaffirmed for me that no matter what you read and what you’ve learned and what kind of lens [which] you view pain, if you can’t connect with people, and if you just feel like you’re not able to amend to the blue collar worker or the CEO white collar person and [be] kind of a shapeshifter on a daily basis, you’re going to struggle regardless of what you know*”—P004.

“*May I say, I think I’ve become even more accepting of the people I work with, the uncertainty of it all, the complexity of it all, the things I do and do not have control over. I think it’s maybe the past year has just accentuated some of the things that I came in with…I have some values that I abide by, and I’ve double downed on a lot of those, so it’s made me feel even more sure about that actually and even more thoughtful with patients and more deliberate with patients about how we go about things*”—P001.

Finally, in asking the question: Can you provide a metaphor for yourself as a pain care provider? Each learner described in detail a metaphor that represented themselves and their role. Two of the metaphors focused on guides or teacher, whereas two focused on metaphors of a figuring out or putting together. Interestingly the metaphors all focused in one way or another on the collaborative and relational dimensions of pain management, and of figuring out the path together with the patient.

“*The role that is hard to play for people is more as a guide. They’re there to keep people safe and try and lead them in a way that they want to and that maybe they can’t always see for themselves first, so I might have to light the way or lead them a little bit and just walking with them. Never carrying them*”—P001.

“*It’s almost a puzzle piece to some extent too. You’re a part of a management plan or you are part of that kind of pain management piece, but I don’t see myself as being the entire piece of the puzzle, and I think it’s sort of working alongside those other people and sort of fitting together what it means within the context of that person’s day-to-day life and sort of how we can support them as best as possible. A puzzle piece photo of a body*”—P002.

“*I guess I look at myself as a teacher because many times I’d say the vast majority of times people haven’t been told what’s going on in any meaningful way…And they haven’t been told what the diagnosis is, what it means, what the structures are involved*”—P003.

“*I’m thinking is like in terms of my role in treating pain. It’s like a puzzle. For me, I want so desperately want the pieces to fit, so it makes a whole picture of what the puzzle is, but the issue is in terms of like this puzzle is sometimes you don’t have the box to reference the photo to*”—P004.

## 4. Discussion

The objective of this study was to explore and understand the lived experience of the first cohort from the MClSc Interprofessional Pain Management of participating in a pain management education program. The study findings highlight the journey of the students’ learning from the reasoning for entering the program; the experience of meaning-making with like-minded colleagues; challenging former understandings of pain management and critical thinking from their previous clinical training; valuing the spirit of interprofessional care including critical thinking, reflection, and creativity while exploring their shift in patient partnerships; and finally working towards becoming competent person-centred pain care providers.

### 4.1. Life-Long Learning

Learners expressed the desire to challenge themselves and the value of life-long learning. According to Tryssenaar and Perkins (2001), continuing competence is a lifelong commitment that all professionals must make [11]. The emphasis in many health curricula of problem solving and lifelong learning are also mirrored in essential competency profiles for license practitioners, as evidence that even at the pre-professional training level the values of continued reflection, challenging assumptions, and continuing professional development are established [11]. In the findings, the majority of participants hinted at, or clearly stated their recognition of the need for continuing education and the value placed on lifelong learning. Reflective practice is recognized as a component of adult learning, and that reflective practice is central lifelong learning as well as professional development in all professions [12]. Participants expressed feeling stale and frustrated with their current offerings from their clinical professions for continuing education. According to Trede (2012), “if the aim is to educate students to become critical, considerate, global citizens and lifelong learners then this should be addressed in all spaces of learning” [13]. As critical thinking and reasoning is one of the five major competencies in the IPM program, learners who participate and complete this program are encouraged to continue to challenge their current understandings of pain management and themselves. The findings also demonstrate that meaning making with like-minded learners to critically challenge theories, ideas, and practices in a space where learners felt comfortable doing so seemed to be a supportive in further developing capabilities for critical reflection and thinking.

### 4.2. Dialogue

In the findings, processes of dialogue featured prominently in participants meaning making. Canadian philosopher, Charles Taylor (1922), claims that people do not acquire the languages needed for self-definition on their own; rather they are introduced to them through exchanges with others who matter to them [14]. Taylor also suggests that the genesis of the human mind is not ‘monological,’ not something each person accomplishes on their own, but dialogical [15]. Participants described how the dialogue with like-minded individuals in stimulating conversations felt important to them. Learners discussed how they made sense of experience and learned through dialogue and reflection with like-minded individuals. Interestingly in relation to the pain field itself van Manen (2016) states, “pain as suffering carries mostly negative connotations in today’s world,” yet “pain forces us to reflect and to give it a place and meaning in our lives” [6].

In, “*Meaning of Pain*”, Buytendijk (1957) acknowledges the complexity of pain and the human body while recognizing pain is not only physical [16]. This program adopted a biopsychosocial approach to pain rather than one single perspective of pain. Buytendijk contends that the human body is a web of meanings each vitally bound to the other and it is not how clinicians react to the body, but rather how the clinican reacts bodily to the meaning of the pain that is important [16]. The thought-provoking conversations and reflective assignments appeared to encourage participants to challenge themselves and others on ideas, practices, and deeper reflection.

### 4.3. Ideal Practice

Another capability that requires practice and was found in the findings to be a component of ideal practice is interprofessional collaboration. While all learners emphasized the need for more interprofessional interactions, some also stressed that the program offered access to a wide variety of experts, and the opportunity for interprofessional collaborations and dialogue. Some participants highlighted the need for mentorship, and the lack of mentorship through their careers. This may be due to the statement that, the progress in interprofessional pain education at the advanced ‘trainee’ stage of learning within the clinical settings has been less than ideal [2], which may be an area to further explore at a later date. In expressing their ideal practice as well as the importance of interprofessional collaboration, the participants made visible their values and biases. Closely related to values is the construction of professional identity. Depoy and Cook Merrill (1988) argue that professional values can be operationalized by being made conscious through articulation, writing, and practice [17]. Taking values into account, professional identity has been defined as the constellation of attributes, beliefs, and values people use to define themselves in specialized, skill-and-education-based occupations or vocations [18].

Given that phenomenology is largely reflective and the MClSc IPM program strongly values reflection while also requiring learners to demonstrate their mastery in the competency of self-awareness and reflection, it important to acknowledge these parallels. It has been suggested that clinical educators challenge themselves with practical concerns regarding the use of reflection and that the goal of educators is to foster dialogue, to be critical through the use of reflective questions, and to facilitate profession professional practice capabilities [19]. Reflective capabilities have been recommended to enhance resilience, healthy professional identity formation, and relationship-centered education [20]. This includes the ability to constructively process emotions and while appreciating multiple perspectives as a basis for empathy. This may also be applied to post-professional learners as reflection is a skill that requires practice. Limitations of this study included that the participants were comprised of four learners from a convenience sample, which may reduce the student sample representation. Future research may include exploring the lived experience of participation in the IPM program with a larger sample size across several cohorts.

## 5. Conclusions

Pain management has been increasingly recognized as an important component of clinical education. To our knowledge, interprofessional pain management has not been rigorously explored especially in a competency-based framework, which we believe to be one of many strengths of our study. So far, findings from the first cohort of indicate that the learners were challenged to think critically, find ways for meaning-making to build confidence to apply to clinical practice, reflect deeply on personal biases, and work to practice reflection-in-action. Many participants also described interprofessional collaboration as a value, and shared examples of the application of knowledge gained from the program to current clinical practice. This program offers a unique approach to learning while creating an online platform to work, collaborate, and challenge like-minded experts in the field of pain. In doing this research, we hope that more practitioners will work towards the goal of becoming a competent person-centred care pain care provider as the participants shared that it is an active process, much like allyship. Ally has become a verb, and in being an ally, the learners of this program expressed a commitment to values of life-long learning and advocacy. Hopefully, the learners of this program will continue to challenge their colleagues, their professions, and the field of pain management itself to think more holistically about pain management.

## Figures and Tables

**Figure 1 healthcare-11-01397-f001:**
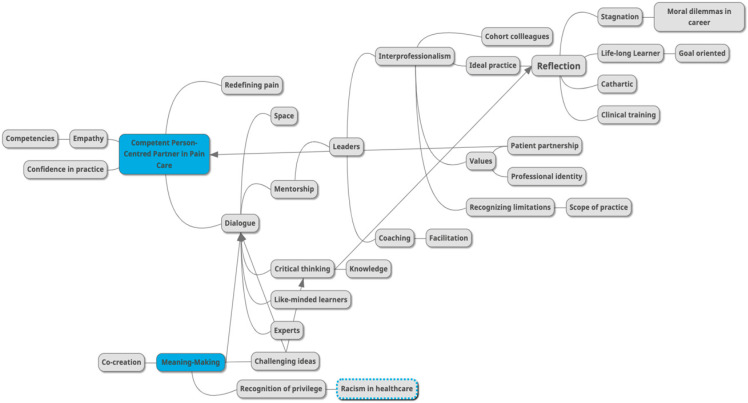
Mind map to organize data and topics of interest.

**Table 1 healthcare-11-01397-t001:** MClSc IPM Competencies and EPAs.

Critical Reasoning and Creativity	Empathic Practice and Reasoning	Self-Awareness and Reflexivity
1.1 Efficiently and effectively searches appropriate sources to find relevant knowledge	2.1 Takes perspective and can subjectively experience another person’s psychological state and intrinsic emotions	3.1 Explores and identifies critical incidents, both personal and professional
1.2 Efficiently and effectively summarizes a peer-reviewed paper	2.2 Identifies and understands a person’s feelings and perspective	3.2 Describes and reflects on cultural and societal biases
1.3 Evaluates new knowledge for trustworthiness and risk bias	2.3 Constructs appropriate responses to convey an understanding of another’s perspective	3.3 Uses reflection to develop a deeper understanding of previous critical incidents
1.4 Interprets the findings of research from a critical social science lens	2.4 Behaves in a non-judgmental, compassionate, tolerant, and empathic way	3.4 Moves beyond reflection and critical thinking to benign introspection
1.5 Synthesizes research evidence with clinical experience	2.5 Tracks changes in the quality of the therapeutic alliance	3.5 Sees self through the eyes of others
1.6 Creates new relevant, defensible, ethically, and socially just knowledge		
Interprofessional Collaboration	Pain Expertise	
4.1 Demonstrates partnership in an interprofessional team	5.1 Describes and interprets pain from a biopsychosocial lens	
4.2 Demonstrates teamwork and collaboration	5.2 Collects information to evaluate and interpret intervention pain	
4.3 Interprets and understands patient health through social determinants of health and social equity	5.3 Appraises, synthesizes, and summarizes the biology of pain	
4.4 Advocates on behalf of patients	5.4 Appraises, synthesizes, and summarizes the psychology of pain	
4.5 Establishes an effective working alliance with patients	5.5 Appraises, synthesizes, and summarizes socio-cultural aspects of pain	
	5.6 Conducts a comprehensive assessment of a patient’s pain experience	
	5.7 Establishes a prognosis/theragnosis for patients in pain	
	5.8 Synthesizes information from assessments to intervention plans	
	5.9 Selects a published model of pain to critically interpret	

Table includes all MClSc Interprofessional Pain Management field competencies and Entrustable Professional Activities.

**Table 2 healthcare-11-01397-t002:** Interview Guide.

Research Questions	Objectives
1. What motivated you to enroll in this program? Prompts: Did you expect this program to impact your practice? How so? What professional or personal experiences led you to applying? What were your hopes in applying to this program? How comfortable did you feel about your competence in pain management prior to applying?	Determine the experience for applying to the new alternative program. Determine the reasoning for applying to the new alternative program. Identify what experiences led the participants to apply. Identify if the participants’ expectations were met in regard to the program.
2. Please describe your experience in the Interprofessional Pain Management program. Prompts: What challenges have you faced during the program? Did you have any transformative experiences during the program? Some people call these ‘ah-ha’ moments. Have you noticed any change in your practice behaviours? Have you stepped outside of your comfort zone? How did that go? What was it like keeping a portfolio?	Interview learners on their lived experience of learning about pain in the MClSc Interprofessional Pain Management (IPM) program. Identify how participants’ understandings of pain has changed. Interview participants to determine if they experienced challenges while enrolled in the IPM program.
3. In what ways have your experiences in the program changed how you think about pain or pain management? Prompts: What were your understandings about pain prior to the program? How did you see yourself as a partner in pain care prior to the program? Have you looked at pain differently? Have your views changed or been challenged? Have you looked at pain management differently? Have your views changed or been challenged? How do you think this experience has shaped your view of pain overall? Are there any examples from your practice you would be willing to share?	Identify participants’ personal beliefs of pain prior to enrolling in the program. Identity current personal beliefs of pain while in the program. Determine if the experience of being in the program has changed participants’ clinical practice. Determine if the experience of being in the program has changed participants’ views of pain overall.
4. In what ways have your experiences in the program shaped how you think about yourself as a pain care provider? Prompts: In what ways, if at all, has this experience shaped how you view yourself as a pain care provider? What core values or beliefs do you hold related to working as a pain care provider? Have these changed? In what ways (if at all) have you changed as a professional as a result of this experience? What do you think constitutes competence in a pain-care provider? What do you think constitutes effectiveness in a pain-care provider?	Determine how participants define their professional identities. Identify if the participants feel that the IPM program will impact their future clinical practice. Determine if participants’ beliefs and assumptions about pain changed since enrolling in the IPM program. In what ways was participation IPM program perceived by participants to shape their professional identity?
5. Can you provide a metaphor for yourself as a pain care provider? Prompts In thinking of a metaphor, have you thought of yourself as something other than a healthcare provider? Some examples others have used to describe themselves and/or a metaphor in similar studies include: juggler, stark trek enterprise captain, karate instructor, a cook, a recipe, the human body…do you relate to any of these? What is your ideal practice? What does that look like? How would your patients and colleagues describe you as a pain-care provider? How would your family and friends describe you as a pain-care provider? Do you feel that you have changed as a clinician over the course of being in this program? If so, how?	Identify any metaphors the participants relate to as pain care providers. Determine participants’ ideal pain care practice. Identify if there are changes in participants’ care since being enrolled in the program. Determine if participants have changed how they view practice as a pain care provider.
6. Do you have any final thoughts you would like to share?	Overall thoughts of the program. Any additional comments to the previous questions asked? Time for participants to ask the interviewer questions.

The interview guide includes questions asked to each participant with additional prompts to engage in dialogue, followed by the objective for each question.

## Data Availability

Data sharing not applicable. No new data were created or analyzed in this study. Data sharing is not applicable to this article.

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
