# Peer review of "Two Puzzles, a Tour Guide, and a Teacher: The First Cohorts’ Lived Experience of Participating in the MClSc Interprofessional Pain Management Program"

_healthcare, 2023, doi:10.3390/healthcare11101397_

Round 1

Reviewer 1 Report (Previous Reviewer 2)

Thank you for these responses and comments. I am happy with the responses to most of the comments and believe the manuscript has been improved, but would like to lift and clarify some of them based on the responses from the authors:

4: I still believe it would be relevant to reference a previously developed model. The biopsychosocial model is one developed by researchers, and while it has become a common term, I still believe it is right to correctly reference the researchers responsible for its development, especially since the model is directly named in the text.

7: I appreciate that rigor is addressed, but I think it would be beneficial to the reader to have a more thorough discussion of what this means: for example member checking, pre-determined “quality” criteria, credibility, transferability, dependability, confirmability, and steps taken to ensure trustworthiness and authenticity of the data and subsequent results.

10: I understand that re-interviewing falls outside the scope of the process as it stands. My suggestion was more along the lines of attempting to interpret the available responses in the dataset provided from the first interviews with reference to these questions. For example, do the responses you received indicate any sense of how the participants interpret their needs in terms of a time-frame – do they discuss their experiences as being a step-wise process in which critical thinking is a necessary first step to be able to fully gain insight into the learning process provided by the program? Taking an interpretivist approach in this way does not require further interviews. I do, however, understand the ethical and methodological limitations which make this a difficult alteration and merely include this comment again in order to communicate my thoughts on this.

12: I do not think it is necessary to ask the participants, but it would help the reader to understand that the authors are transparent and aware of the possibility of biases, so simply adding a sentence about possible limitations may be beneficial. I still also believe that the authors would benefit from adding a strengths section in order to show other benefits of this research.

Thank you again for this interesting study, and I appreciate the responses to my comments!

Author Response

Reviewer 2 Report (Previous Reviewer 3)

The evaluation processes aim to support the improvement of the quality of training, teaching and the pedagogical development of the institution, the teams and individual teachers. They aim to better understand the perception of students concerning study conditions, training and teaching. 

The evaluation process presented in this publication is well adapted to pain management.

Author Response

Thank you very much, Reviewer 2 for your thoughtful feedback and taking the time to review this manuscript. 

This manuscript is a resubmission of an earlier submission. The following is a list of the peer review reports and author responses from that submission.

Round 1

Reviewer 1 Report

In recent years, the phenomenology of Husserl and Heidegger has been re-approved and attempts have been made to apply it to sociological analysis, and it seems to have become a standard method as hermeneutic phenomenology.

Although these qualitative research methods tend to be artisanal, it is good that objectivity seems to have been added to them with the help of recent advances in computers and It is understood that the authors have conducted a proper analysis based on this method.

Though It differs from the classical method consisting of hypothesis and testing, and is difficult to evaluate in some respects, it may be important to introduce these new methods of analysis. Regardless of the importance of the results, I agree that the paper is publishable as a presentation of a new analytical attempt.

Reviewer 2 Report

Thank you for the opportunity to review this interesting study. I will begin with a short summary of how I interpret the paper, followed by specific comments. This is a phenomenological interview study in which participants provided their lived experiences of participation in a Master’s program on pain management. Results indicate that they believe they needed to get out of a stagnant environment, received social support from like-minded colleagues, learned critical thinking skills, developed a more holistic approach, and reflected on becoming more competent in person-centered care. The authors conclude that the Master’s program seems to have achieved the learning goals and is a feasible method of creating more allies in the field of pain management.

Overall, this is a very interesting and relevant study. I believe it is important to take this holistic approach, and the results presented in this manuscript seem to be another step in the right direction. I have some concerns, however, about the methodology and presentation of the data and results. I believe that transparency and rigor need to be addressed more thoroughly, especially in regards to the process of identifying themes during data analysis. I also have concerns with the chosen method, as the results are presented in a factual and descriptive manner, despite the authors writing that the analysis is based on an interpretivist approach.

Below are some more specific comments and suggestions:

Abstract:

1. Lines 19-21 – I interpret this as being the “methods” section, and think it might benefit from a short description of how the lived experiences were uncovered in relation to the research question (i.e. a short description of coding processes or theme and subtheme generation). As you have already described the phenomenon in question in your aims, I don’t think the second half of this sentence adds much to the abstract.

2. Lines 25-28 – I feel that this is somewhat of a restatement of the results, and could be improved by relating to clinical practice or future research (i.e. how can these results help us?)

3. Abstract overall – Is the numbering part of the journal formatting? This was slightly confusing, and I found myself wondering if “(2)” was a reference before noticing the other numbers corresponding to different sections of the abstract.

Introduction:

4. Line 65 – I believe it would be appropriate here to reference the biopsychosocial model (i.e. Brewer, Andersen, and van Raalte, 2002). I am also not sure I understand how reference number 5 fits in with this sentence in terms of the biopsychosocial approach.

Methods:

5. In the methods section, I would encourage the authors to refer to the ISSM COREQ checklist. Specifically items 5 (in relation to interview studies), 11, 13-16, 18, 20, 22-25, 27, and 28, which I was not able to easily find in the text.

6. Line 151 – The authors have used the term “identify” several times. I understand the difference in methodologies, but what is described seems to be more of an active, interpretivist, and reflexive role in data analysis. I therefore wonder if the argument from Braun and Clarke (Braun V, Clarke V. One size fits all? What counts as quality practice in (reflexive) thematic analysis? Qual Res Psychol. 2021;18(3):328-352) is a relevant one to bring up in regards to the active role of the researcher – that is “generating” rather than “identifying.” If the approach is in fact interpretivist, the results as they are presented below do not fit with this approach, as they are descriptive, indicating a passive role from the researcher.

7. Methods Overall: I believe it would be beneficial to the reader if there was more of a discussion regarding rigor in the data analysis process, especially in order to establish a sense of trustworthiness. For example, Smith and McGannon have an article in which this is discussed nicely (Smith B, McGannon KR. Developing rigor in qualitative research: Problems and opportunities within sport and exercise psychology. Int Rev Sport Exerc Psychol. 2018;11(1):101-121). I believe it would also benefit the reader to more clearly describe the process of identifying or generating themes, as it is not very clear from the text how this was done. (See also my comment 1).

Results:

8. The analysis and themes seem to be well thought-out and there seems to be a clear and interesting thread throughout, with good support from the provided quotations.

9. I feel that the results are simply a statement of the themes. Based on the methodology used (“Intepretive” phenomenology), I was expecting more of an interpretation of the results. Smith JA. Evaluating the contribution of interpretative phenomenological analysis. Health Psychol Rev. 2011;5(1):9-27 provides an interesting checklist (pg 17) on how to ensure quality in an IPA study

10. There does seem to be a more clearly defined interpretivist approach in the presentation of responses to the question regarding the metaphor. I think this style of interpretation throughout the results would be a good addition to the themes identified earlier. This might allow a more thorough understanding of how they experienced it, rather than just re-stating their words. I think it would be very interesting to know the authors’ interpretations of how the participants became competent person-centered care providers, for example. Is it simply gaining knowledge and/or experience, or do they interpret it as being a holistic and temporal view in which critical thinking needs to be developed first? Do the participants feel that the program provided this, or is this a natural progression of their professional lives?

As Smith writes on page 24 of his article on IPA: “the researcher is engaging in the double hermeneutic: trying to make sense of the participant and trying to making sense of their experience.” (Smith JA. Evaluating the contribution of interpretative phenomenological analysis. Health Psychol Rev. 2011;5(1):9-27.)

Discussion:

11. I find the discussion relevant and interesting.

12. Line 569 – Here the authors introduce a limitation; however, I believe it might be relevant to lift other limitations. For example, is there a possibility that the interview guide was in some way leading or presumptive? The question “in what ways have your experiences changed how you think” may presume that there has indeed been a change. This may especially be true given the role of the interviewer in the program, potentially leading to interviewer or social desirability biases. Likewise, I think it would be relevant to add a strengths section in order to lift the fact that this provides a much deeper and nuanced discussion compared to quantitative research, and as such has the potential to be used to improve the program in many ways.

Overall Comments

13. After having read the whole manuscript, I believe it would be beneficial to have more information on the program itself in the introduction. The core competencies are presented, but not the methods of achieving these learning goals. I think this would help to put the results in context. I am thinking specifically of the results from Theme 2, in which participants discuss expert and small group discussion. It was not clear from the introduction that these were included, so it was somewhat difficult to understand the context of the program and its potential benefits in relation to the lived experiences of the participants.

14. I would recommend looking over the entirety of the manuscript for “readability.” The sentences are very long in some places, making it somewhat difficult to follow (e.g. Lines 37-42, 65-69, etc.). There are also places where extra (e.g. Lines 26 and 155, etc.) or missing (e.g. line 16 and 110, etc.) words, and incorrect pluralization (e.g. Lines 18, 21, and 40, etc.) make comprehension somewhat difficult.

15. As a side note, I would encourage the authors to address the text in the “Acknowledgements” section on lines 602-604.

Again thank you for this interesting manuscript!

Reviewer 3 Report

Pain, and especially spinal pain, is one of the areas that anatomical-physiology is not sufficient to describe, explain and, a fortiori, to treat. More than the technique, it is the health professional himself who will determine the final result. The evaluation and adaptation of training therefore requires an interpretative paradigm of the changes that occur in the health professional.

The process will be useful for teachers and trainers.

On the level of the title, the metaphor of the two puzzles would deserve to be made explicit in the text.

The time between the Zoom interview and the end of the training is not specified. Is there a follow-up interview to monitor the practice of the course?